# Evaluating Parental Knowledge and Behaviors Regarding Developmental Toxicants in Jazan, Saudi Arabia Using the Prevention of Toxic Chemicals in the Environment for Children Tool (PRoTECT)

**DOI:** 10.3390/healthcare12171764

**Published:** 2024-09-04

**Authors:** Ahmad Y. Alqassim

**Affiliations:** Family and Community Medicine Department, Faculty of Medicine, Jazan University, Jazan 45142, Saudi Arabia; aalqassim@jazanu.edu.sa

**Keywords:** developmental toxicants, parental awareness, PRoTECT, environmental health, neurodevelopmental disorders, public health policy, toxicant exposure prevention, Saudi Arabia, Jazan region

## Abstract

This study evaluated the level of knowledge among parents in Jazan, Saudi Arabia, regarding substances that can harm child development. The Prevention of Toxic Chemicals in the Environment for Children Tool (PRoTECT) was used for this assessment. A cross-sectional survey using a multi-stage cluster random sampling approach was undertaken among 424 parents who were enlisted from eight primary healthcare centers (PHCCs). The PRoTECT score’s median value was 72 out of 90, suggesting a generally high level of awareness. The study found that individuals with higher education, particularly those with postgraduate degrees, had greater awareness of protecting their children’s health. Interestingly, unemployed individuals and those residing in mountainous areas also demonstrated higher awareness, possibly due to having more time to focus on their children’s health and well-being. Most participants (68.2%) acknowledged the correlation between exposure to toxic chemicals during pregnancy and early childhood, and the subsequent development of neurodevelopmental disorders. The study found a solid foundation of knowledge, with 85.1% of participants interested in learning more about reducing children’s exposure, but it also stressed the need for specific actions to turn awareness into prevention. These findings would help policymakers develop effective strategies, such as targeted educational campaigns, collaboration with healthcare providers, utilization of media channels, and encouragement of community-led initiatives, to reduce children’s exposure to developmental toxicants in line with national and global environmental health initiatives. Future research should focus on longitudinal consciousness and behavior evaluations and regional environmental contaminants.

## 1. Introduction

Neurodevelopmental disorders, including autism spectrum disorder (ASD), attention deficit hyperactivity disorder (ADHD), and learning difficulties, have become increasingly prevalent in recent decades, affecting millions of children worldwide [1,2,3]. Beyond these specific disorders, exposure to developmental toxicants has also been linked to a range of adverse health outcomes, such as respiratory problems, endocrine disruption, and cancer [4]. While genetic factors contribute to these disorders and health issues, environmental toxicants, especially during prenatal and early childhood, are crucial to neurodevelopmental trajectories and overall health [5,6]. Recent studies showed a 0.26–0.36% prevalence for ASD, 12.4% for ADHD, and 23.89% for learning difficulties in Saudi Arabia, putting a strain on the healthcare system and society [7,8,9]. Genetic factors contribute to these disorders, but environmental toxicants, especially during prenatal and early childhood, are crucial to neurodevelopmental trajectories [5,6]. These disorders, along with other health problems associated with toxicant exposure, can present challenges, but early identification and intervention can significantly improve outcomes and quality of life for affected individuals and their families [10]. In addition to environmental pollutants such as heavy metals, pesticides, and air pollutants, harmful substances from food, beverages, and indoor pollutants can also contribute to the rise in developmental disorders and adverse health outcomes [11,12,13]. It is estimated that these disorders costs USD 64.8 billion annually in the United States [14]. Understanding and mitigating the role of environmental toxicants in the etiology of developmental disorders and other health problems is a global public health priority due to their long-term effects on education, employment, health, and social functioning [15].

Given the significant impact of environmental toxicants on neurodevelopmental disorders, understanding and addressing parental awareness is crucial in mitigating children’s exposure. Parents play a crucial role in minimizing children’s exposure to environmental toxicants, as they make daily decisions about their children’s environment, diet, and product use [16]. Parental awareness and understanding of environmental toxicants can significantly affect children’s exposure [17]. Studies across various countries have revealed differing levels of parental awareness about environmental toxins. In the United States, studies showed low concern among pregnant women about chemicals such as phthalates, Bisphenol A (BPA), and mercury [18]. European research also found misconceptions about chemical bioaccumulation, fetal/child impacts, exposure sources, and avoidance strategies [19]. Studies have shown that socioeconomic and educational factors affect awareness in developing countries [20].

In Saudi Arabia, particularly in regions like Jazan, there is a significant knowledge gap regarding parental awareness of environmental toxicants [21]. Effective interventions tailored to the local cultural context and environmental challenges require understanding local awareness levels. Saudi Arabia has implemented various public health campaigns and policies to reduce children’s exposure to developmental toxicants, such as the National Environmental Strategy [22] and the Saudi Food and Drug Authority’s (SFDA) regulations on pesticide residues in food [23]. However, the effectiveness of these programs has been hindered by the lack of region-specific data on parental awareness and understanding of environmental health risks. This knowledge gap limits the ability to develop targeted interventions and risk communication strategies that resonate with local communities. This knowledge gap hinders Saudi Arabia’s targeted public health campaigns and policies to reduce children’s developmental toxicant exposure [24]. In light of the Project TENDR consensus statement [6], which emphasizes the long-term effects of these exposures on child development and health, assessing and addressing parental awareness in specific regional contexts is not just important, but essential for protecting children’s health and development. To address this knowledge gap, tools such as the Prevention of Toxic Chemicals in the Environment for Children (PRoTECT) questionnaire has been developed to assess parental awareness and understanding of environmental toxicants [25]. PRoTECT is an 18-item validated questionnaire designed to evaluate parents’ knowledge about toxic chemicals and brain development, level of concern about toxic chemicals, and preferences for prevention of neurodevelopmental disorders. It uses a 5-point Likert scale, and covers domains such as pregnancy complications, expected dangers to children, required public policies, health promotion, and protection from toxic chemicals. While PRoTECT is relatively new, it has shown good internal consistency and content validity in its initial validation study [25]. Its main benefits include comprehensive assessment of parental awareness and attitudes, while limitations may include potential social desirability bias in responses.

About 1.5 million people live in the Jazan region, a rapidly developing region in southwest Saudi Arabia on the Red Sea [26]. The Jazan region has a unique environmental profile due to its coastal, agricultural, and mountainous landscape. Industrial activities, intensive agriculture, and urban development may expose the region to environmental toxins [21]. Industrialization, intensive agriculture, and urbanization may pollute the region [21]. Pesticides from extensive agricultural practices [27], heavy metals and air pollutants from industrial emissions, especially around Jazan Economic City [28], and potential water contaminants from rapid urbanization and inadequate waste management, are of concern [29,30]. The widespread agricultural practices in Jazan, involving intensive pesticide use, present a significant risk of toxic exposure, particularly for children [27]. Rapid industrialization, including Jazan Economic City, may increase air and water pollution [28]. The Environmental Law and its implementing regulations have been implemented in Saudi Arabia, but their effectiveness in addressing developmental toxicant exposures, particularly in Jazan, is not assessed [31]. Despite limited initiatives targeting developmental toxicants in children, the Saudi Vision 2030 prioritizes environmental sustainability and public health [32]. Region-specific research is needed to inform public health strategies in Jazan due to its unique environmental challenges and potential for increased toxicant exposures. This research gap restricts the development of effective interventions to reduce children’s developmental toxicant exposure in this unique geographical and cultural context [21,24].

The current study addresses the knowledge gap in parental awareness of developmental toxicants in Jazan, a region with unique environmental challenges and a lack of region-specific research. The validated PRoTECT questionnaire was used to assess parental environmental toxicant awareness and understanding [25]. The research also explored demographic awareness, knowledge, perception, and preventative behavior. The findings can identify knowledge gaps and misconceptions to support Saudi Vision 2030’s environmental sustainability and public health goals through targeted awareness programs [2]. The study contributes to the global understanding of parental awareness of developmental toxicants in diverse cultural and environmental contexts. It examines developmental toxicant knowledge, perceptions, and behaviors among parents and childbearing adults in Jazan, Saudi Arabia.

## 2. Materials and Methods

### 2.1. Study Design and Setting

This analytical cross-sectional study was conducted in Jazan, Saudi Arabia. Jazan, in southwest coastal Saudi Arabia, has 1.5 million residents [33]. The Jazan region has a unique environmental profile due to its coastal, agricultural, and mountainous landscape [21]. Geographic diversity and rapid industrialization and urbanization complicate environmental toxicant exposures.

### 2.2. Sampling and Participant Recruitment

The study utilized a multi-stage cluster random sampling technique to select participants from the Jazan region. Jazan includes seven health sectors: Central, North, South, Middle, Mountain, Bani Malik, and West. To ensure a representative sample, four sectors were randomly chosen for the study (Appendix A). Two primary Healthcare centers (s) were randomly selected from each chosen sector, resulting in a total of eight participating primary healthcare centers (PHCCs) (Appendix A). This method facilitated a diverse representation of the population across various geographic regions within Jazan.

The sample size was determined using the formula: n = [Z^2^P(1 − P)]/d^2^, where n represents the sample size, Z is the standard normal variate (1.96 for a 95% confidence level), P is the estimated proportion of the population (set at 0.5 to maximize the sample size), and d is the desired level of precision (0.05) [34]. The computation resulted in an initial sample size of 383. To accommodate possible non-responses, the researchers augmented the sample size by 10%, aiming for 420 participants. The final sample comprised 424 participants, slightly surpassing the intended goal to accommodate both parents equally (Appendix A).

Participants were considered eligible for inclusion if they met the following criteria: (1) being 18 years of age or older, (2) being parents, (3) attending one of the participating PHCCs, and (4) expressing willingness to take part in the study. The employed sampling strategy and eligibility criteria ensured a diverse and representative sample of parents in the Jazan region, enabling a thorough evaluation of their awareness regarding developmental toxicants [20].

### 2.3. Data Collection Tool

The study employed the Prevention of Toxic Chemicals in the Environment for Children (PRoTECT) questionnaire, a validated instrument to evaluate parental knowledge and comprehension of environmental toxic substances [25]. The PRoTECT questionnaire was specifically developed to assess the community’s understanding and preferences concerning hazardous substances and their potential influence on the cognitive development of children. This tool was chosen for its capacity to capture various aspects of parental awareness, such as knowledge, attitudes, and behaviors concerning developmental toxicants.

The original English version of the PRoTECT questionnaire underwent a meticulous translation process to guarantee its suitability in the Saudi Arabian context. Two autonomous bilingual specialists conducted the initial translation from English to Arabic, which was subsequently followed by reverse translation to English by two distinct bilingual specialists. The resolution of any inconsistencies was achieved through consensus discussions, which included the research team and a group of experts specializing in environmental health and pediatrics. The objective of this process was to preserve the conceptual equivalence of the questionnaire while also ensuring its cultural appropriateness. The Arabic translation was subsequently tested with 25 participants from the intended audience to validate its cultural appropriateness, comprehensibility, and dependability. The feedback obtained from the pilot test was utilized to further refine the questionnaire, thereby improving its appropriateness for the Jazan community.

The ultimate questionnaire comprised two primary sections. The initial section collected demographic data using 17 variables, encompassing age, gender, educational attainment, income, and family structure. The second section, derived from the PRoTECT questionnaire, consisted of 18 items that evaluated the level of understanding regarding the effects of toxic chemicals on children’s health [25]. The items were categorized into five subscales: pregnancy complications (consisting of 3 items), expected dangers to children (consisting of 3 items), required public policies (consisting of 4 items), health promotion (consisting of 3 items), and protection from toxic chemicals in children (consisting of 5 items). Participants provided responses using a 5-point Likert scale that ranged from “strongly disagree” to “strongly agree”. This framework facilitated a thorough assessment of parental comprehension and perspectives on substances that can harm development, encompassing various factors such as individual awareness and preferences regarding regulations [16,25].

### 2.4. Data Collection Procedure

A comprehensive training program was implemented to ensure consistent data collection practices across all PHCCs. The principal investigator convened meetings with a representative from each PHCC, including either physicians, nurses, or officers. The sessions covered the study objectives, elucidated the data collection mechanism, and offered practical training with the digital questionnaire platform. The objective of this approach was to reduce the influence of interviewer bias and ensure uniformity in the procedures used to collect data. The data collection period extended from January to July 2024, providing ample time to achieve the desired sample size while considering potential difficulties in recruiting participants.

The data collection process involved the use of a self-administered questionnaire, which was made available through Google Forms. Participants could access the questionnaire either by scanning a QR code or by using a specific URL. The decision to adopt this digital method was made to improve the precision of data and streamline the process of gathering information. To ensure inclusivity in the study population, trained data collectors were available to administer the questionnaire on behalf of participants who did not have access to smartphones or preferred assistance.

Various strategies were put in place to guarantee the accuracy and entirety of the data. The digital questionnaire incorporated embedded logic checks to mitigate inconsistent responses and minimize data omissions [35]. Systematic surveillance of incoming data was carried out to detect any consistent problems or trends of missing responses. If any questionnaires were found to be incomplete, participants were promptly contacted within 48 h to provide the missing information, if feasible. The researchers meticulously recorded and examined instances of non-responses to evaluate the possibility of non-response bias. Questionnaires that had more than 20% missing data were not included in the final analysis in order to ensure the accuracy and reliability of the data, in accordance with established guidelines in survey research [36].

### 2.5. Statistical Analysis

The statistical analyses were conducted using IBM SPSS Statistics software version 27 (IBM Corp. in Armonk, NY, USA). Statistical measures were computed for all variables. Categorical variables were summarized using frequencies and percentages, while continuous variables were computed using means and standard deviations. The normality of the PRoTECT scores was evaluated by conducting the Shapiro–Wilk test and visually examining the histograms. Due to the presence of a non-normal distribution of scores, as indicated by a skewness coefficient of −1.39 and a Shapiro–Wilk test result of 0.90 with a *p*-value less than 0.001, non-parametric tests were used to compare groups. Kruskal–Wallis tests were employed to compare PRoTECT scores among categories comprising more than two groups, such as age groups, educational levels, and working sectors. Mann–Whitney U tests were utilized to compare two groups, such as gender and occupation in the health sector. The PRoTECT questionnaire’s internal consistency was evaluated by calculating Cronbach’s alpha for each subscale and the overall questionnaire. A multiple linear regression analysis was performed to determine the factors associated with PRoTECT scores. The variables that demonstrated significant associations in the bivariate analyses were incorporated into the model. The model provided the regression coefficients, standard errors, t-values, and *p*-values for each variable. A *p*-value less than 0.05 was deemed statistically significant for all statistical tests. The analysis took into consideration the intricate sampling design, which involved clustering at the PHCC level.

## 3. Results

### 3.1. Demographic and Socioeconomic Background Characteristics of the Study Participants

Table 1 presents the demographic and socioeconomic characteristics of the 424 study participants. The sample was equally divided between fathers and mothers (50% each). Thirty-four percent of participants were between the ages of 36 and 45, and thirty-two percent were between the ages of 26 and 35. Most participants (63.7%) had a university education, and 92.7% were married. In terms of employment, 27.1% of people were unemployed and 57.5% worked in the public sector. Seventy percent of the population had three or more children. Most people lived in towns (56.4%), followed by villages (32.8%). Sixty-seven percent of monthly income fell between 5001 and 15,000 Saudi Riyal (SAR). Remarkably, 53.1% of participants had a family member employed in the health sector, and 28.3% of participants were employed in the field. 5.4% of respondents said their children had learning disabilities, 3.8% had autism, and 10.1% had attention deficit hyperactivity disorder.

### 3.2. Parental Knowledge and Awareness Regarding Developmental Toxicants

The results of parental awareness regarding the impact of toxic chemicals on children are presented in Table 2. In total, 68.2% of the participants concurred that minimizing exposure to hazardous chemicals during pregnancy and early childhood could potentially decrease the likelihood of developing disorders such as ADHD or autism. A total of 72.2% of the participants recognized that being exposed to toxic chemicals while pregnant could elevate the likelihood of experiencing developmental disorders. Significantly, 80.2% of participants acknowledged that toxic chemicals have a greater detrimental impact on children and infants compared to adults. Regarding public policies, a majority of 73.1% of participants expressed confidence in the existence of effective legislation that guarantees the absence of harmful levels of toxic chemicals in food and personal care products. Approximately 80.9% of respondents expressed their support for improving programs and policies aimed at preventing the presence of harmful chemicals in consumer products that can negatively impact children. The subscales exhibited good internal consistency, as evidenced by Cronbach’s alpha values ranging from 0.729 to 0.890.

Participants’ awareness of and attitudes toward shielding kids from hazardous chemicals are shown in Table 3. The majority of respondents (81.6%) concurred that medical professionals ought to tell them about hazardous substances endangering the health of their family members. The majority of participants (85.1%) said they would like to know more about lowering the amount of hazardous chemicals exposed to kids. Remarkably, 76.7% of participants expressed confidence in scientific data pertaining to the health consequences of hazardous substances. Although 73.8% of participants thought all parents had an equal chance to shield their kids from harmful substances, a larger portion (83.7%) said they would make an effort to limit kids’ exposure if they knew how. A total of 82.3 percent of respondents said they try to purchase products free of hazardous chemicals, and 81.1 percent said they are concerned about the chemicals that their family may be exposed to. It is interesting to note that 65.6% of respondents said they believed most businesses produced goods free of dangerously high amounts of hazardous chemicals. This scale’s Cronbach’s alpha was 0.875 to 0.891, which denotes strong internal consistency.

### 3.3. The Prevention of Toxic Chemicals in the Environment for Children Tool (PRoTECT) Total Scores

The distribution of the Prevention of Toxic Chemicals in the Environment for Children Tool (PRoTECT) total scores is presented in Figure 1. The distribution of the scores is skewed [Skewness coefficient = −1.39; Shapiro–Wilk = 0.90, *p* < 0.001]. The Box Plot in Figure 2 showed that the PRoTECT scores range from 18 to 90, with a median of 72 and IQR of 11 (Q3 = 78 − Q1 = 67). Higher scores indicate good awareness about the Prevention of Toxic Chemicals in the Environment for Children among the study participants, and almost 106 (25%) of study participants scored more than 78 marks. In contrast, the same percentage scored marks less than 67.

The distribution of PRoTECT scores across different socio-demographic attributes is shown in Table 4. The median PRoTECT score for mothers and fathers was 72. With the exception of individuals 56 years of age and older, whose median score was marginally higher at 75, the median scores for all age groups were consistent at 72. A postgraduate degree holder scored the highest (median = 77), while someone with only a primary education scored the lowest (median = 66). Educational level demonstrated significant differences (*p* = 0.031). Significant differences in place of residence were also observed (*p* = 0.023), with residents of mountainous areas scoring higher (median = 73) than residents of villages and towns (median = 72 for both). Having a family member working in the medical field caused a significant difference (*p* = 0.038), but employment in the field did not (*p* = 0.151). It is interesting to note that there were no appreciable differences in scores between participants who had children with learning disabilities, autism, or ADHD and those who did not. The population appears to have a relatively uniform level of awareness, as evidenced by the consistency of median scores across many categories. Education and place of residence appear to be significant determinants of awareness.

### 3.4. Factors Associated with Parental Knowledge Regarding Developmental Toxicants

Table 5 presents the results of a multiple linear regression analysis examining factors associated with PRoTECT awareness scores. Education level was found to be a strong indicator, as all levels demonstrated higher scores in comparison to the reference group (primary education or lower). The results showed that postgraduate education had the highest positive correlation (β = 16.21, *p* = 0.001), followed by intermediate (β = 12.21, *p* = 0.007), secondary (β = 9.43, *p* = 0.008), and university education (β = 9.04, *p* = 0.010). The participants’ place of residence had a significant impact on their scores. Those from mountainous areas scored higher than those from villages (β = 6.11, *p* = 0.004). Another important factor that influenced the results was the employment status of the participants. Non-working individuals had higher scores compared to those who were employed in government jobs (β = 4.08, *p* = 0.023). The income level demonstrated a notable correlation solely within the SAR 10,001 to 15,000 category, with a coefficient of 4.46 and a *p*-value of 0.028. Curiously, individuals who did not have a family member working in the health sector had higher scores, with a beta coefficient of 2.93 and a *p*-value of 0.020. There were no statistically significant associations between PRoTECT scores and gender, age groups, marital status, number of children, or personal employment in the health sector.

## 4. Discussion

This study provides the first comprehensive assessment of parental awareness regarding developmental toxicants in Jazan, Saudi Arabia. Parents are generally aware, with a median PRoTECT score of 72 out of 90. Interestingly, 68.2% of participants recognized the link between toxic chemical exposure during pregnancy and early childhood and ADHD or autism. The level of awareness is comparable to and sometimes higher than that reported in studies from developed countries. In the US, less than half of pregnant women worried about chemical exposures [17]. Several factors unique to Jazan, such as rapid industrialization and urbanization [21], may have raised public awareness about environmental health risks. The study analysis found that that higher education, mountainous residence, and, intriguingly, unemployment, were associated with higher awareness. The positive relationship between education and awareness supports global studies [20], emphasizing the importance of education in enhancing environmental health risk awareness. Higher awareness among mountainous residents may be related to exposure to toxic chemicals like pesticides used during agricultural practices, highlighting the influence of local environmental contexts on risk perception. The unexpected finding that unemployed individuals demonstrated higher awareness (β = 4.08, *p* = 0.023) warrants further investigation. One possible explanation is that these groups may have more time to focus on their children’s health and well-being, as individuals with more free time may be able to devote more attention to health-related matters [13]. However, this interpretation requires additional research to confirm.

The high level of parental awareness observed in this study must be understood within the unique environmental and cultural context of Jazan. The region’s rapid industrialization, exemplified by projects like Jazan Economic City [28], has likely heightened public consciousness about environmental health risks. At the same time, the high occurrence of intensive agricultural practices, which involves the heavy use of pesticides, might have played a role in raising awareness about chemical exposures [27]. The coexistence of traditional practices and modern development in Jazan results in a multifaceted landscape of risk perception. The process of urban development and its related challenges, such as the possibility of water contamination due to insufficient waste management, worsens these environmental concerns. In addition, the enactment of the Environmental Law and its regulations in Saudi Arabia [31] may have contributed to an increase in public awareness. However, it is still unknown how effective these measures are in addressing the specific issue of developmental toxicant exposures in Jazan. The Saudi Vision 2030, which prioritizes environmental sustainability and public health [32], is likely influencing public discussions and raising awareness about these matters. The combination of fast-paced progress, customary methods, and changing government regulations results in a distinct environment for parents to be conscious of developmental toxins in Jazan. The study’s findings indicate that the population has a heightened level of awareness, which can be attributed to the environmental impacts of rapid changes. This awareness is influenced by both local experiences and national initiatives.

The findings of this study have significant implications for public health strategies and policies in Jazan and Saudi Arabia at large. The significant level of parental awareness observed (68.2%) establishes a solid basis for focused public health interventions. Nevertheless, the differences in awareness among various demographic groups indicate the necessity for customized strategies. An example of this is the significant correlation between the level of education and awareness (β = 16.21, *p* = 0.001 for postgraduate education), which emphasizes the need to incorporate environmental health education at different levels of the educational system. This is in line with the goals of Saudi Vision 2030, which prioritizes the development of human resources and the preservation of the environment [32]. The study found that residents of mountainous areas have a higher level of awareness (β = 6.11, *p* = 0.004), indicating that implementing environmental health initiatives targeted specifically to these regions could be highly effective. Existing awareness can be utilized by public health campaigns to promote targeted preventive behaviors and emphasize the significance of minimizing exposure to developmental toxicants. Furthermore, the surprising discovery of increased consciousness among individuals who are unemployed (β = 4.08, *p* = 0.023) emphasizes the possibility of implementing community-oriented educational initiatives that can effectively engage various sectors of the population. According to the Environmental Law [31], policymakers should enhance regulations regarding the use of pesticides, industrial emissions, and waste management. This is especially important in rapidly developing regions such as Jazan Economic City [28]. The high level of interest (85.1%) in learning more about reducing children’s exposure to harmful substances indicates a receptive audience for environmental health initiatives. To raise public awareness, policymakers should develop targeted educational campaigns for high-risk regions and demographic groups, collaborate with healthcare providers, utilize media channels, and encourage community-led initiatives. Implementing these recommendations can help capitalize on the existing awareness and interest among parents in Jazan to drive positive change. To effectively contribute to the broader goals of environmental sustainability and improved public health outlined in Saudi Vision 2030 [32], public health strategies should focus on addressing specific concerns and utilizing the existing awareness in Jazan.

While this study provides valuable insights into parental awareness of developmental toxicants in Jazan, it also highlights several areas for future research. Firstly, comprehensive examinations of particular environmental toxicants commonly found in Jazan, such as pesticides utilized in agricultural practices [27] and industrial discharges from Jazan Economic City [28], are necessary to gain a better understanding of local exposure hazards. Furthermore, conducting longitudinal studies to assess the efficacy of awareness campaigns in modifying parental behaviors and mitigating children’s exposure to toxicants would be highly beneficial. This information could be used to guide the development of focused interventions that are in line with the public health objectives of Saudi Vision 2030 [32]. Furthermore, it is essential to conduct research that investigates the correlation between levels of awareness and the implementation of preventive behaviors. This is because having a high level of awareness does not necessarily result in taking appropriate action [17]. While the current study found high levels of parental intention to reduce children’s toxic exposures and self-reported protective purchasing behaviors, future research could benefit from more detailed assessments of specific actions taken and the barriers and facilitators to implementing these behaviors. Furthermore, conducting research on the heightened awareness observed in unemployed individuals could offer valuable insights into the development of effective strategies for disseminating information. Ultimately, conducting comparative studies across various regions of Saudi Arabia could assist in customizing national policies to suit specific local circumstances, considering the observed regional variations. These research directions would enhance our understanding of environmental health risks in Jazan and facilitate evidence-based policy-making to minimize children’s exposure to developmental toxicants.

### Strengths and Limitations

This study possesses various strengths, notably the utilization of the validated PRoTECT questionnaire [25], which augments the dependability and comparability of our findings. The substantial sample size (*N* = 424) and comprehensive representation across Jazan’s health sectors establish a strong basis for our conclusions. Moreover, our emphasis on Jazan’s distinct environmental circumstances [21] provides valuable observations on parental consciousness in swiftly evolving areas with particular environmental difficulties. However, some limitations should be noted. The use of a cross-sectional design in this study restricts our ability to make causal conclusions, and the sampling method employed may have introduced selection bias, which could result in an overrepresentation of health-conscious individuals who visit PHCCs. The responses may have been influenced by social desirability bias, especially when it comes to awareness and preventive behaviors [36]. Furthermore, although PRoTECT is extensive, it may not encompass all intricacies of local environmental issues. The accuracy of the data may be influenced by the self-reported nature, particularly when it comes to neurodevelopmental disorders in children. Finally, the analysis is based on the Likert scale, which has a limitation. Different respondents may interpret the response options differently, especially the “neutral” mid-points, leading to inconsistent interpretations of the scale.

## 5. Conclusions

This study offers significant findings regarding parental knowledge of developmental toxicants in Jazan, Saudi Arabia. It reveals that there is generally a high level of awareness among parents, with a median PRoTECT score of 72 out of 90 and 68.2% of participants recognizing the link between toxic chemical exposure during pregnancy/early childhood and neurodevelopmental disorders. The significant correlations between education, place of residence, and level of awareness emphasize the necessity for customized interventions. The findings indicate a strong base of knowledge, with 85.1% of participants interested in learning more about reducing children’s exposure, but also emphasize the need for focused public health efforts, especially in converting awareness into preventive actions. Unforeseen outcomes, such as increased consciousness among individuals who are jobless, justify the need for additional investigation. These insights are essential for policymakers to develop effective strategies to reduce children’s exposure to toxic substances, targeting high-risk regions and demographic groups, in line with national and global initiatives. Future research should prioritize conducting longitudinal assessments to evaluate the long-term effects of awareness and behavior change. Additionally, there is a need for investigations that specifically target local environmental toxicants. This study enhances the worldwide comprehension of environmental health awareness in swiftly progressing regions and establishes a basis for constructing healthier environments for children in Jazan and comparable areas globally.

## Figures and Tables

**Figure 1 healthcare-12-01764-f001:**
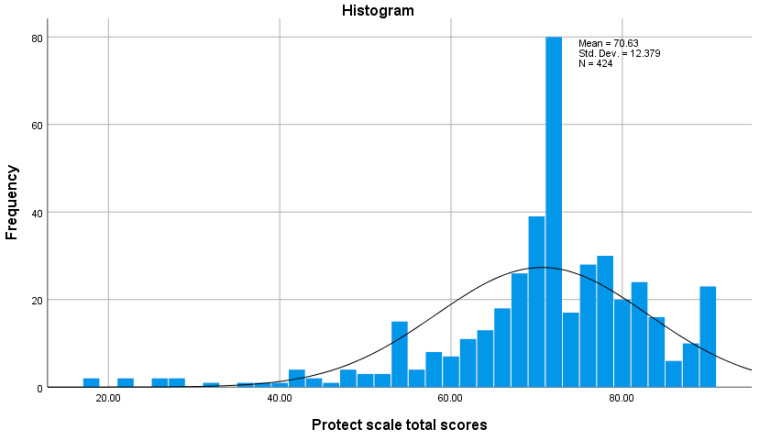
The distribution of Prevention of Toxic Chemicals in the Environment for Children Tool (PRoTECT) total scores.

**Figure 2 healthcare-12-01764-f002:**
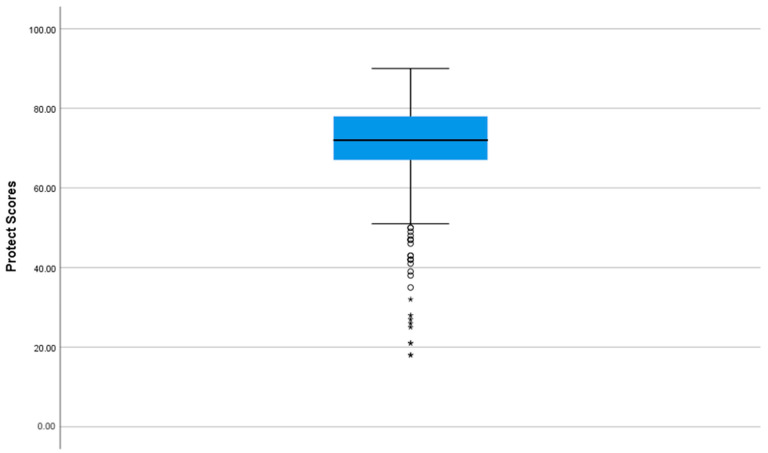
Box Plot of Prevention of Toxic Chemicals in the Environment for Children Tool (PRoTECT) scores. The circle (o) represents a mild outlier. The asterisk (*) represents an extreme outlier.

**Table 1 healthcare-12-01764-t001:** Demographic and socioeconomic background characteristics of the study participants according to gender (*N* = 424).

Variables		All *N* = 424	Fathers *N* = 212	Mothers *N* = 212
		*N* (%)	*N* (%)	*N* (%)
Age groups (years)	18–25	55 (13.0)	20 (9.4)	35 (16.5)
	26–35	145 (34.2)	66 (31.1)	79 (37.3)
	36–45	159 (37.5)	84 (39.6)	75 (35.4)
	46–55	58 (13.7)	37 (17.5)	21 (9.9)
	56 and more	7 (1.7)	5 (2.4)	2 (0.9)
Educational level	Primary or less	15 (3.5)	6 (2.8)	9 (4.2)
	Intermediate	15 (3.5)	7 (3.3)	8 (3.8)
	Secondary	105 (24.8)	56 (26.4)	49 (23.1)
	University	270 (63.7)	131 (61.8)	139 (65.6)
	Postgraduate	19 (4.5)	12 (5.7)	7 (3.3)
Marital status	Married	393 (92.7)	201 (94.8)	192 (90.6)
	Divorced	20 (4.7)	8 (3.8)	12 (5.7)
	Widowed	11 (2.6)	3 (1.4)	8 (3.8)
Working sector	Government	244 (57.5)	157 (74.1)	87 (41.0)
	Private	34 (8.0)	23 (10.8)	11 (5.2)
	Other	31 (7.3)	14 (6.6)	17 (8.0)
	Not Working	115 (27.1)	18 (8.5)	97 (45.8)
Children ever born	1–2 Children	127 (30.0)	65 (30.7)	62 (29.2)
	3–4 Children	174 (41.0)	93 (43.9)	81 (38.2)
	Five and more children	123 (29.0)	54 (25.5)	69 (32.5)
Place of residence	Village	139 (32.8)	69 (32.5)	70 (33.0)
	Twon	239 (56.4)	117 (55.2)	122 (57.5)
	Mountain	46 (10.8)	26 (12.3)	20 (9.4)
Monthly income	5000 SAR or less	73 (17.2)	31 (14.6)	42 (19.8)
	5001 to 10,000 SAR *	144 (34.0)	68 (32.1)	76 (35.8)
	10,001 to 15,000 SAR	143 (33.7)	75 (35.4)	68 (32.1)
	15,001 to 20,000 SAR	41 (9.7)	24 (11.3)	17 (8.0)
	More than 25,000 SAR	23 (5.4)	14 (6.6)	9 (4.2)
Do you work in the health sector?	Yes	120 (28.3)	59 (27.8)	61 (28.8)
	No	304 (71.7)	153 (72.2)	151 (71.2)
Does any other family member work in the health sector?	Yes	225 (53.1)	96 (45.3)	129 (60.8)
	No	199 (46.9)	116 (54.7)	83 (39.2)
Do you have children who suffer from learning difficulties?	Yes	23 (5.4)	11 (5.2)	12 (5.7)
	No	387 (91.3)	189 (89.2)	198 (93.4)
	Not sure	14 (3.3)	12 (5.7)	2 (0.9)
Do you have children who suffer from autism?	Yes	16 (3.8)	12 (5.7)	4 (1.9)
	No	397 (93.6)	192 (90.6)	205 (96.7)
	Not sure	11 (2.6)	8 (3.8)	3 (1.4)
Do you have children who suffer from attention deficit hyperactivity disorder?	Yes	43 (10.1)	24 (11.3)	19 (9.0)
	No	362 (85.4)	177 (83.5)	185 (87.3)
	Not sure	19 (4.5)	11 (5.2)	8 (3.8)

* USD 1 = 3.75 Saudi Riyal (SAR).

**Table 2 healthcare-12-01764-t002:** Parental awareness of the impact of toxic chemicals on children, focusing on pregnancy complications, expected dangers, and required public policies (*N* = 424).

Scale	Items ^$^	Disagree * *N* (%)	Neutral *N* (%)	Agree ^#^ *N* (%)	Mean (SD)	Cronbach’s Alpha
Pregnancy Complications	Reducing exposure to toxic chemicals during pregnancy and early childhood can help reduce the risk of a child developing disorders such as ADHD or autism.	52 (12.3)	83 (19.6)	289 (68.2)	3.81 (1.1)	0.729
	Exposure to toxic chemicals during pregnancy can increase the risk of a child developing developmental disorders.	50 (11.8)	68 (16.0)	306 (72.2)	3.84 (1.1)	
	Pregnant women are highly exposed to toxic chemicals.	79 (18.6)	114 (26.9)	231 (54.5)	3.43 (1.2)	
Expected Dangers on Children	The daily toxic chemicals in our lives, such as air pollution or lead in drinking water, can increase the risk of a child developing conditions like ADHD or autism.	49 (11.6)	92 (21.7)	283 (66.7)	3.70 (1.0)	0.832
	Toxic chemicals are generally more harmful to children and infants than to adults.	34 (8.0)	50 (11.8)	340 (80.2)	4.00 (1.0)	
	Children will benefit more from regulating and reducing toxic chemicals (preventing injury) in developmental disorders than from (treating) these conditions.	34 (8.0)	76 (17.9)	314 (74.1)	3.88 (0.9)	
Required Public Policies	Most countries in the world invest similar amounts in preventing developmental conditions as they do in treating these conditions.	24 (5.70)	111 (26.2)	289 (68.2)	3.80 (0.9)	0.890
	We have effective legislation to ensure that food and personal care products do not contain harmful levels of toxic chemicals.	23 (5.4)	91 (21.5)	310 (73.1)	3.90 (0.9)	
	When it comes to addressing developmental disorders affecting children, countries spend money managing and treating these conditions. I believe that more should be spent researching ways to prevent children from suffering from these conditions.	23 (5.4)	73 (17.2)	328 (77.4)	3.97 (0.9)	
	We should enhance our programs and policies to ensure that consumer products do not contain the toxic chemicals that harm our children.	25 (5.9)	56 (13.2)	343 (80.9)	4.05 (0.9)	

^$^ Items (1–10) of the Prevention of Toxic chemicals in the Environment for Children Tool (PRoTECT); SD = standard deviation; ^#^ agree involves strongly agree and somewhat agree responses; * disagree includes strongly disagree and somewhat disagree. The median and mode = 4 for all table items.

**Table 3 healthcare-12-01764-t003:** Participants’ awareness and attitudes regarding protection of children from toxic chemicals (*N* = 424).

Scale	Items ^$^	Disagree * *N* (%)	Neutral *N* (%)	Agree ^#^ *N* (%)	Mean (SD)	Cronbach’s Alpha
Health Promotion	If toxic chemicals threatened my family’s health, my doctor or healthcare provider should inform me about it.	23 (5.4)	55 (13.0)	346 (81.6)	4.05 (0.9)	0.891
	I am interested in learning more about how to reduce children’s exposure to toxic chemicals.	19 (4.5)	44 (10.4)	361 (85.1)	4.15 (0.9)	
	Among all sources of information on the health effects of toxic chemicals, I trust the information provided by the scientists who study them.	18 (6.1)	51 (17.2)	227 (76.7)	4.07 (0.8)	
Protection from Toxic Chemicals in Children	All parents have equal opportunities to protect their children from toxic chemicals such as pesticides or heavy metals, regardless of income level, race and origin, or where they live.	39 (9.2)	72 (17.0)	313 (73.8)	3.88 (1.0)	0.875
	If I knew how to reduce children’s exposure to toxic chemicals, I would definitely try to do so.	21 (5.0)	48 (11.3)	355 (83.7)	4.15 (0.9)	
	I try to buy products that do not contain toxic chemicals that could harm my family.	23 (5.4)	52 (12.3)	349 (82.3)	4.12 (0.9)	
	I am concerned that my family is being exposed to toxic chemicals.	26 (6.1)	54 (12.7)	344 (81.1)	4.07 (0.9)	
	I trust that most companies manufacture products that do not contain harmful levels of toxic chemicals.	41 (9.7)	105 (24.8)	278 (65.6)	3.73 (0.9)	

^$^ Items (10–18) of the Prevention of Toxic chemicals in the Environment for Children Tool (PRoTECT) questionnaire; SD = standard deviation; ^#^ agree involves strongly agree and somewhat agree responses; * disagree includes strongly disagree and somewhat disagree. The median and mode = 4 for all table items.

**Table 4 healthcare-12-01764-t004:** Distribution of PRoTECT scores across socio-demographic characteristics (*N* = 424).

Variables		Median	IQR		*p* Value ^#^
			Q1	Q3	
Gender	Father	72	67	78	0.599 *
	Mother	72	67	78	
Age groups (years)	18–25	72	63	78	0.872
	26–35	72	67	77	
	36–45	72	67	78	
	46–55	72	65	78	
	56 and more	75	70	77	
Educational level	Primary or less	66	54	72	0.031
	Intermediate	75	71	82	
	Secondary	72	68	78	
	University	72	67	78	
	Postgraduate	77	69	83	
Marital status	Married	72	67	78	0.738
	Divorced	76	58	80	
	Widowed	72	71	76	
Working sector	Government	72	66	78	0.485
	Private	72	66	75	
	Other	73	64	80	
	Not Working	72	67	79	
Children ever born	1–2 Children	72	65	80	0.676
	3–4 Children	72	68	77	
	Five and more children	72	64	78	
Place of residence	Village	72	65	79	0.023
	Town	72	66	77	
	Mountain	73	70	85	
Monthly income	5000 SAR or less	72	62	75	0.552
	5001 to 10,000 SAR	72	66	79	
	10,001 to 15,000 SAR	72	68	78	
	15,001 to 20,000 SAR	72	68	78	
	More than 25,000 SAR	72	63	81	
Do you work in the health sector?	Yes	72	67	80	0.151 *
	No	72	67	78	
Does any other family member work in the health sector?	Yes	72	66	77	0.038 *
	No	72	68	79	
Do you have children who suffer from learning difficulties?	Yes	69	60	75	0.101
	No	72	68	78	
	Not sure	68	54	87	
Do you have children who suffer from autism?	Yes	76	64	81	0.879
	No	72	67	78	
	Not sure	70	54	90	
Do you have children who suffer from attention deficit hyperactivity disorder?	Yes	71	63	79	0.237
	No	72	68	78	
	Not sure	67	58	77	

^#^ *p*-value based on Kruskal–Wallis; * *p*-value based on Mann–Whitney U.

**Table 5 healthcare-12-01764-t005:** Multiple linear regression analysis of factors associated with PRoTECT awareness scores (*N* = 424).

Variables		Coef	SE Coef	T-Value	*p* Value
Gender	Father (ref)				
	Mother	0.09	1.38	0.07	0.947
Age groups (years)	18–25 (ref)				
	26–35	−0.52	2.04	−0.25	0.799
	36–45	0.77	2.19	0.35	0.723
	46–55	−0.22	2.54	−0.09	0.930
	56 and more	6.96	5.34	1.30	0.193
Educational level	Primary or less(ref)				
	Intermediate	12.21	4.53	2.70	0.007
	Secondary	9.43	3.51	2.69	0.008
	University	9.04	3.48	2.59	0.010
	Postgraduate	16.21	4.63	3.50	0.001
Working sector	Government (ref)				
	Private	0.29	2.41	0.12	0.906
	Other	3.71	2.46	1.51	0.132
	Not Working	4.08	1.79	2.28	0.023
Monthly income	5000 SAR or less(ref)				
	5001 to 10,000 SAR *	1.85	1.89	0.98	0.327
	10,001 to 15,000 SAR	4.46	2.02	2.21	0.028
	15,001 to 20,000 SAR	2.54	2.72	0.93	0.352
	More than 25,000 SAR	1.55	3.30	0.47	0.639
Place of residence	Village (ref)				
	Twon	−0.11	1.32	−0.09	0.932
	Mountain	6.11	2.11	2.89	0.004
Marital status	Married (ref)				
	Divorced	2.36	2.91	0.81	0.418
	Widowed	1.30	3.86	0.34	0.735
Children ever born	1–2 Children (Ref)				
	3–4 Children	0.52	1.55	0.34	0.735
	Five and more children	−2.21	1.83	−1.21	0.228
Do you work in the health sector?	Yes(ref)				
	No	−1.05	1.45	−0.73	0.467
Does any other family member work in the health sector?	Yes (ref)				
	No	2.93	1.26	2.34	0.020

Coef = Regression coefficient; SE Coef = Standard error of the coefficient; ref = reference category; * USD 1 = 3.75 Saudi Riyal (SAR).

## Data Availability

The data presented in this study are available on request from the corresponding author.

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
