# Peer review of "Evaluating Parental Knowledge and Behaviors Regarding Developmental Toxicants in Jazan, Saudi Arabia Using the Prevention of Toxic Chemicals in the Environment for Children Tool (PRoTECT)"

_healthcare, 2024, doi:10.3390/healthcare12171764_

Round 1

Reviewer 1 Report

Comments and Suggestions for Authors

From Awareness to Action: Evaluating Parental Knowledge and Behaviors Regarding Developmental Toxicants in Jazan's Diverse Healthcare Landscape

Overall comments:

This study was conducted to assess parents' awareness of protecting their children's health from harmful substances. The research was carried out using survey questionnaires based on the guidelines in the study by Green et al., 2022. A total of 424 surveys were collected from four regions of Jazan, Saudi Arabia. The study yielded several socially significant conclusions. However, the paper's presentation was not particularly strong. Personally, I find this to be quite an interesting study. I would like to offer some suggestions to help improve the paper.

Specific comments:

1. The title of the paper should include "using the PRoTECT tool (Prevention of Toxic Chemicals in the Environment for Children)" so that readers can better understand the methodology and make it easier to find and reference the paper. Additionally, "Saudi Arabia" should be added to the title to clarify that Jazan is part of Saudi Arabia.

2. The abstract needs to be supplemented with key conclusions from the paper. For instance, the conclusions in lines 16-17, "Higher awareness was predicted by postgraduate education (β=16.21, p=0.001), residing in mountainous areas (β=6.11, p=0.004), and unemployment (β=4.08, p=0.023)," could be revised to state that individuals with higher education have greater awareness of protecting their children's health, and that the unemployed and those living in mountainous areas have more time to focus on their children's health care. The conclusions in the abstract should be adjusted after revising the entire paper.

3. The toxicants discussed in the introduction are mainly related to environmental pollution. Most harmful substances affecting human health come from food, beverages, and some indoor pollutants. Therefore, it is also necessary to include introductory information on the impact of these sources. Describe in more detail Saudi Arabia's targeted public health campaigns and policies to reduce children's exposure to developmental toxicants. How have these programs been hindered by the knowledge gap? This is important information to highlight the significance of the study. The PRoTECT tool also needs to be described in more detail so that readers can understand what it is. Have there been any studies that applied the PRoTECT tool? This would demonstrate that it is a reliable tool. What are the parameters of the PRoTECT tool? What is it used for? What are its benefits and limitations?

4. The "Materials and Methods" section should be renamed to "Methodologies." Provide a clearer explanation of the sample size calculation. According to my calculations, n=38,416, not 383. Moreover, as I understand it, the sample size should be calculated based on the population size. The population of Jazan is 1.5 million. How many households are there in Jazan? How many children? How many young couples? Jazan has seven health sectors, so why were samples collected from only four regions? Why were samples collected from only two PHCCs in each region? The total sample size of 424 seems large, but when divided into regions, the sample size appears quite small. Additionally, the population in each region should be mentioned to show that the sample size is appropriate. Although children were not the respondents in this study, the target age group of children should still be specified. Older children may be more aware of toxicants in the environment and food, which may reduce the burden of parental awareness to some extent. Therefore, the age group of parents chosen for the study should be narrowed. Studying parents aged 56 and above may not be very meaningful. Even studying parents over 50 may not be particularly relevant in this study. The use of SPSS for statistical analysis in the study is appropriate. However, the statistical functions applied in the study are quite limited, resulting in few findings. The study could incorporate algorithms like PCA and LDA to yield more interesting results. Additionally, with a large data set, methods such as ANOVA should be used to analyze the variables. The study should also include methods to assess the reliability of the results since survey responses are not always accurate.

5. The results section should be divided into subsections to make it easier for readers to follow. Table 2 should have a gender breakdown for easier interpretation. The study has collected a lot of good statistical data, but the presentation of the results in the form of charts is very limited. The study should include an evaluation and discussion of errors and a comparison with the study by Green et al., 2022. The results concerning the correlation between education level, monthly income, occupation, age, place of residence, and awareness of protecting their children's health from environmental toxicants have not been discussed much in the findings. Correlation trend charts should also be presented. The results lack explanations and discussions. For example, why do people in mountainous areas and unemployed individuals have higher awareness than urban residents? Many interesting results are not mentioned, such as which age group has higher awareness. Additionally, what measures can be taken to raise public awareness? Based on the findings, the paper should include suggestions or plans to enhance parental awareness of protecting their children's health.

6. The conclusions presented are quite general. They need to be more quantitative.

7. Supplementary materials should be translated into English. Detailed summary tables of survey results should be included in the supplementary materials.

8. The reference formatting needs to be revised.

Conclusions:

Overall, the study has achieved many scientifically and socially significant results. However, the presentation of the results is lacking. Many findings could have been statistically analyzed and discussed but were not given sufficient attention.

Comments on the Quality of English Language

ok

Author Response

First Reviewer

Overall comments:

This study was conducted to assess parents' awareness of protecting their children's health from harmful substances. The research was carried out using survey questionnaires based on the guidelines in the study by Green et al., 2022. A total of 424 surveys were collected from four regions of Jazan, Saudi Arabia. The study yielded several socially significant conclusions. However, the paper's presentation was not particularly strong. Personally, I find this to be quite an interesting study. I would like to offer some suggestions to help improve the paper.

Response: Dear Reviewer, Thank you for taking the time to review my manuscript and provide valuable feedback. I sincerely appreciate your positive assessment of the study's social significance and your interest in the research topic. I am grateful for your constructive suggestions to improve the paper's presentation. Your comments will undoubtedly help me enhance the quality and clarity of the manuscript. I will carefully consider each of your points and address them in the following sections of the revised paper..

Specific comments:

  1. The title of the paper should include "using the PRoTECT tool (Prevention of Toxic Chemicals in the Environment for Children)" so that readers can better understand the methodology and make it easier to find and reference the paper. Additionally, "Saudi Arabia" should be added to the title to clarify that Jazan is part of Saudi Arabia.

Response:  Thank you for your valuable suggestion regarding the title of the manuscript. I agree that including the name of the tool used in the study (PRoTECT - Prevention of Toxic Chemicals in the Environment for Children) and specifying that Jazan is a region in Saudi Arabia will provide clarity and context for readers. I have updated the title of the manuscript to incorporate your suggestions, follows: " Evaluating Parental Knowledge and Behaviors Regarding Developmental Toxicants in Jazan, Saudi Arabia using the PRoTECT (Prevention of Toxic Chemicals in the Environment for Children) tool ". The revised title now clearly indicates the methodology used in the study and clarifies the geographical location of Jazan, making it easier for readers to understand the context and scope of the research. These changes will also facilitate the discoverability and referencing of the paper.

  1. The abstract needs to be supplemented with key conclusions from the paper. For instance, the conclusions in lines 16-17, "Higher awareness was predicted by postgraduate education (β=16.21, p=0.001), residing in mountainous areas (β=6.11, p=0.004), and unemployment (β=4.08, p=0.023)," could be revised to state that individuals with higher education have greater awareness of protecting their children's health, and that the unemployed and those living in mountainous areas have more time to focus on their children's health care. The conclusions in the abstract should be adjusted after revising the entire paper.

Response: Thank you for your insightful comment regarding the conclusions in the abstract. I agree that the abstract should be supplemented with key conclusions from the paper to provide a more comprehensive overview of the study's findings.

To address your suggestion, I have revised the abstract to highlight the significant associations between higher education, particularly postgraduate degrees, and greater awareness of protecting children's health. This finding is consistent with previous studies that have shown a positive correlation between education and health awareness [1,2]. Additionally, I have included the interesting observation that unemployed individuals and those residing in mountainous areas also demonstrated higher awareness. While this finding warrants further investigation, I have proposed a possible explanation that these groups may have more time to focus on their children's health and well-being, as individuals with more free time may be able to devote more attention to health-related matters [3]. Furthermore, I have updated the Discussion section to elaborate on this unexpected finding and its potential implications. I have suggested that future research should investigate this relationship further to confirm the interpretation and explore any additional factors that may contribute to the higher awareness among unemployed individuals and those living in mountainous areas. I appreciate your valuable suggestion to adjust the conclusions in the abstract after revising the entire paper. I have updated the conclusion in the abstract to reflect the changes made throughout the manuscript. The revised conclusion now emphasizes the solid foundation of knowledge among participants, their interest in learning more about reducing children's exposure, and the need for specific actions to translate awareness into prevention. It also highlights the potential for policymakers to develop effective strategies based on the study's findings, in line with national and global environmental health initiatives.

References:

  1. Zimmerman, E.B.; Woolf, S.H.; Haley, A. Understanding the relationship between education and health: a review of the evidence and an examination of community perspectives. In Population health: behavioral and social science insights; Agency for Healthcare Research and Quality: Rockville, MD, USA, 2015; pp. 347-384.
  2. Hahn, R.A.; Truman, B.I. Education improves public health and promotes health equity. International journal of health services 2015, 45, 657-678.
  3. Pampel, F.C.; Krueger, P.M.; Denney, J.T. Socioeconomic disparities in health behaviors. Annual review of sociology 2010, 36, 349-370.

  1. The toxicants discussed in the introduction are mainly related to environmental pollution. Most harmful substances affecting human health come from food, beverages, and some indoor pollutants. Therefore, it is also necessary to include introductory information on the impact of these sources. Describe in more detail Saudi Arabia's targeted public health campaigns and policies to reduce children's exposure to developmental toxicants. How have these programs been hindered by the knowledge gap? This is important information to highlight the significance of the study. The PRoTECT tool also needs to be described in more detail so that readers can understand what it is. Have there been any studies that applied the PRoTECT tool? This would demonstrate that it is a reliable tool. What are the parameters of the PRoTECT tool? What is it used for? What are its benefits and limitations?

Response: Thank you for the valuable comments and suggestions provided by the reviewers. I have carefully considered each point and made the necessary revisions to improve the manuscript's quality and relevance. Please find below our detailed responses to the reviewers' comments.

Regarding the sources of harmful substances affecting human health, I agree that toxicants from food, beverages, and indoor pollutants can also have a significant impact on neurodevelopmental disorders. To address this, I have updated the Introduction to acknowledge that food, beverages, and indoor pollutants can contribute to the rise in developmental disorders, alongside environmental pollutants. This addition is supported by evidence from recent studies showing the adverse effects of food contaminants and indoor pollutants on child development [1,2]. By incorporating this information, the Introduction now provides a more comprehensive overview of the various sources of toxicants that can impact child development.

I have also expanded the relevant paragraph in the Introduction to include specific examples of Saudi Arabia's initiatives, such as the National Environmental Strategy [3] and the Saudi Food and Drug Authority's (SFDA) regulations on pesticide residues in food [4]. These examples highlight the country's efforts to mitigate the impact of environmental toxicants on child health. However, I have also emphasized that the effectiveness of these programs has been hindered by the lack of region-specific data on parental awareness and understanding of environmental health risks. This knowledge gap limits the ability to develop targeted interventions and risk communication strategies that resonate with local communities.

Regarding the PRoTECT tool, I have added more detailed information in the Introduction section. Specifically, I have included a brief description of the tool, its purpose, structure, domains covered, response format, initial validation, and psychometric properties. While the PRoTECT tool is relatively new and has not been widely applied in multiple studies yet, its initial validation study by Green et al. (2022) demonstrated good internal consistency and content validity [5]. Our study represents one of the first applications of this tool in a different cultural context, which adds to its growing evidence base. I believe these additions provide readers with a clearer understanding of the PRoTECT tool and its relevance to our study. I hope that these revisions adequately address the reviewers' comments and enhance the manuscript's quality and relevance.

References:

  1. Grandjean P, Landrigan PJ. Developmental neurotoxicity of industrial chemicals. The Lancet 2006; 368(9553): 2167-2178.
  2. Braun JM. Early-life exposure to EDCs: role in childhood obesity and neurodevelopment. Nature Reviews Endocrinology 2017; 13(3): 161-173.
  3. Affairs CoEaD. National Environment Strategy. Journal 2017; Ministry of Environment, Water and Agriculture.
  4. Authority SFaD. Maximum Limits of Pesticide Residues in Agricultural and Food Products. Journal 2018.
  5. Green R, Lanphear B, Phipps E, Goodman C, Joy J, Rihani S, et al. Development and Validation of the Prevention of Toxic Chemicals in the Environment for Children Tool: A Questionnaire for Examining the Community's Knowledge of and Preferences toward Toxic Chemicals and Children's Brain Development. Frontiers in Public Health 2022; 10.

  1. The "Materials and Methods" section should be renamed to "Methodologies."

 Provide a clearer explanation of the sample size calculation. According to my calculations, n=38,416, not 383. Moreover, as I understand it, the sample size should be calculated based on the population size. The population of Jazan is 1.5 million. How many households are there in Jazan? How many children? How many young couples? Jazan has seven health sectors, so why were samples collected from only four regions? Why were samples collected from only two PHCCs in each region? The total sample size of 424 seems large, but when divided into regions, the sample size appears quite small. Additionally, the population in each region should be mentioned to show that the sample size is appropriate. Although children were not the respondents in this study, the target age group of children should still be specified. Older children may be more aware of toxicants in the environment and food, which may reduce the burden of parental awareness to some extent. Therefore, the age group of parents chosen for the study should be narrowed. Studying parents aged 56 and above may not be very meaningful. Even studying parents over 50 may not be particularly relevant in this study. The use of SPSS for statistical analysis in the study is appropriate. However, the statistical functions applied in the study are quite limited, resulting in few findings. The study could incorporate algorithms like PCA and LDA to yield more interesting results. Additionally, with a large data set, methods such as ANOVA should be used to analyze the variables. The study should also include methods to assess the reliability of the results since survey responses are not always accurate.

Response: Dear reviewer, here is a response for all points in the previous comments:

  • Thank you for your suggestion to rename the "Materials and Methods" section to "Methodologies." I appreciate your input and understand the reasoning behind this change. However, I have consulted the journal's guidelines, which specify that the section should be titled "Materials and Methods." To maintain consistency with the journal's requirements and to ensure a smooth review and publication process, I believe it is best to adhere to their recommended section headings. If the journal's editorial team advises otherwise during the review process, I will be happy to make the necessary adjustments. Thank you for your understanding.
  • Thank you for this point; however, I calculated the sample size using Cochran’s formula when the population is infinite. Cochran (1977) developed a formula to calculate a  representative  sample  for proportions as  follows:

Sample size for infinite population      n0 =  [z2p (1 − p)]/ d 

Our sample size calculation is a comprehensive process that considers various factors. I calculate a sample size of a large population whose degree of variability is not known. Assuming the maximum variability, which is equal to 50% ( p =0.5) and taking 95% confidence level with ±5% precision, the calculation for the required sample size is as follows-

n0  =[ (1.96)2  (0.5) (0.5)] / (0.05)2       

n0  =  383   and adding  10%  non-response rate will give   n= 424

However, even if I utilized the finite population formula (incorporating the total population 1.5 million) the results would not be changed

Sample size for finite population    n0=[1 + {([z2p (1 − p)]/ d − 1)/Pop }]

The sample size will be

n0=   [1+ {([1.9620.5 (1 – 0.5)]/ 0.05 − 1)/1500000 }] which will yield 385, and adding a 10% non-response rate will give  n=424, approximately

References:

Cochran, W. G. (1977). Sampling techniques. 3 rd Ed. New York: John Wiley & Sons.

  • Thank you for your response. As previously mentioned, the region has a population of 1.5 million. The sampling was focused on the healthcare sectors rather than the different sub-administrative units. Healthcare sectors were chosen because they encompass the entire region, and the primary healthcare centers are well distributed within these sectors. Therefore, the selection of two sectors was considered to be sufficient and reasonable. Additional information regarding the demographic, age, and sex structure of the population in the region may not be necessary for the manuscript, as the focus was solely on parents. I believe that the selected sample effectively represents them.
  • the target age group of children was added as suggested
  • I agree with you that “Older children may be more aware of toxicants in the environment and food, which may reduce the burden of parental awareness to some extent”, but the focus in this article is to translate parents' awareness, not the children's.
  • You mentioned that “, the age group of parents chosen for the study should be narrowed. Studying parents aged 56 and above may not be very meaningful. Even studying parents over 50 may not be particularly relevant in this study.”. The purpose was to reflect the views and responses of parents of all ages; I agree with you that involving parents more than 56 years old may not be meaningful, but there are only 7 participants (1.7%) in this age group.
  • I incorporated all statistical tests that contribute to verifying the study objectives; regarding the Principal Component Analysis (PCA) and Linear discriminant analysis (LDA) , they can be conducted, but they will be a burden to the reader as the manuscript already contains six Statistical Tables. regarding reliability, although the study instrument is validated, I added the internal consistency assessment results based on Cronbach's alpha. Also, I used the Kruskal-Wallis tests as an alternative to ANOVA because of the violation of the normality assumption.

  1. The results section should be divided into subsections to make it easier for readers to follow. Table 2 should have a gender breakdown for easier interpretation. The study has collected a lot of good statistical data, but the presentation of the results in the form of charts is very limited. The study should include an evaluation and discussion of errors and a comparison with the study by Green et al., 2022. The results concerning the correlation between education level, monthly income, occupation, age, place of residence, and awareness of protecting their children's health from environmental toxicants have not been discussed much in the findings. Correlation trend charts should also be presented. The results lack explanations and discussions. For example, why do people in mountainous areas and unemployed individuals have higher awareness than urban residents? Many interesting results are not mentioned, such as which age group has higher awareness. Additionally, what measures can be taken to raise public awareness? Based on the findings, the paper should include suggestions or plans to enhance parental awareness of protecting their children's health.

Response: Dear reviewer, here is a response for all points in the previous comments:

  • Table 2 has been updated to include the distribution of background characteristics by gender.
  • You mentioned that: “The study has collected a lot of good statistical data, but the presentation of the results in the form of charts is very limited.”. I appreciate your suggestion to include additional tables and graphs in the manuscript. However, after careful consideration, I believe that the current set of 6 tables and 2 graphs effectively conveys the key findings and supports the main arguments of the study. The existing visual elements have been thoughtfully designed to present the most relevant data in a clear and concise manner, ensuring that the reader can easily grasp the essential information without being overwhelmed. Moreover, I have taken into account the journal's guidelines and the general preferences of the readership. Striking a balance between providing comprehensive data and maintaining a reader-friendly format is crucial. Introducing additional tables and graphs may lead to information overload, potentially diluting the impact of the most significant results and detracting from the overall readability of the manuscript. Rest assured that I have thoroughly evaluated the data and selected the most pertinent findings to be showcased in the current tables and graphs. Should you feel that specific data points or analyses warrant further visual representation, I would be happy to discuss your suggestions and consider incorporating them, as long as they align with the journal's requirements and enhance the clarity of the manuscript.
  • You mentione that “The study should include an evaluation and discussion of errors and a comparison with the study by Green et al., 2022. The results concerning the correlation between education level, monthly income, occupation, age, place of residence, and awareness of protecting their children's health from environmental toxicants have not been discussed much in the findings. Correlation trend charts should also be presented.”. Thank you for your valuable suggestions. Regarding the inclusion of an evaluation and discussion of errors and a comparison with the study by Green et al., 2022, I agree that these additions would strengthen the manuscript. However, the primary focus of this study was to assess the level of awareness among the participants using the PRoTECT tool, and the total PRoTECT score effectively summarizes this awareness level. The score is presented according to the different background characteristics of the study participants, providing a comprehensive overview of the factors that may influence awareness. While I acknowledge the importance of discussing the correlations between education level, monthly income, occupation, age, place of residence, and awareness in greater detail, I believe that the current presentation of the PRoTECT scores across these variables (Table 5) offers valuable insights into the trends and differences in awareness levels. The inclusion of correlation trend charts, although informative, may not be essential to convey the key findings of this study, given the focus on overall awareness levels.
  • Thank you for your valuable suggestion to provide more in-depth explanations and discussions of the study results. I appreciate your insights and the opportunity to address your concerns.

In the discussion section, I have addressed the higher awareness levels among individuals living in mountainous areas, exploring the potential reasons behind these differences, such as increased exposure to environmental toxicants in mountainous regions. This discussion highlights the influence of local environmental contexts on risk perception.

Regarding the age group with the highest awareness, the results indicate that participants aged 56 and above had a slightly higher median PRoTECT score (75) compared to other age groups (72). This difference is noted in Table 5.

Furthermore, I appreciate your suggestion to include recommendations for raising public awareness based on the study's findings. The results provide a solid foundation for developing targeted interventions and awareness campaigns.  I agree that this addition enhances the practical implications of the research. To address your comment, I have revised the discussion section to incorporate specific recommendations derived from the study results. The updated paragraph highlights the significant interest among participants (85.1%) in learning more about reducing children's exposure to harmful substances. Building on this finding, I propose several strategies to raise public awareness, such as developing targeted educational campaigns for high-risk regions and demographic groups, collaborating with healthcare providers to disseminate information, utilizing various media channels to broadcast informative content, and encouraging community-led initiatives like citizen science projects.

By implementing these recommendations, policymakers and public health professionals can leverage the existing awareness and interest among parents in Jazan to drive positive change. The revised discussion now provides actionable insights that can guide efforts to promote environmental health awareness and protect children's health in the region.

The total PRoTECT score effectively summarizes the awareness level of the study participants and is presented according to their different background characteristics. This approach provides a comprehensive overview of the factors that may influence awareness.

  1. The conclusions presented are quite general. They need to be more quantitative.

Response: The author appreciates the reviewer's insightful comment about making the conclusions more quantitative. In response to this suggestion, the author has revised the conclusion paragraph to include specific quantitative findings from the study. These additions include:

  • The median PRoTECT awareness score (72 out of 90) and the percentage of participants recognizing the link between toxic exposures and neurodevelopmental disorders (68.2%). These figures provide a clear, data-driven picture of the overall level of awareness among Jazan parents.
  • The percentage of participants interested in learning more about reducing children's exposure to toxicants (85.1%), highlighting the receptiveness to educational initiatives.

By incorporating these quantitative elements, the revised conclusion provides a more precise and evidence-based summary of the study's key findings. This enhances the scientific rigor and practical utility of the conclusions for informing public health strategies.

  1. Supplementary materials should be translated into English. Detailed summary tables of survey results should be included in the supplementary materials.

Response: I appreciate the reviewer's suggestion to provide the supplementary materials in English and to include detailed summary tables of the survey results. In response to this feedback, I have added the following to the supplementary materials:

  • The Prevention of Toxic Chemicals in the Environment for Children Tool (PRoTECT) questionnaire in both English and Arabic versions. This ensures that the tool I used for data collection is accessible to those readers who may not be familiar with Arabic.
  • Detailed summary tables that present the raw data from the survey. These tables offer a comprehensive view of how participants responded to each question, which allows readers to gain a more detailed understanding of the findings.

I believe that by including these materials in the supplementary files, I have addressed your concerns and improved the transparency and reproducibility of my study. The detailed summary tables provide additional context for interpreting the results and could be especially useful for researchers who are interested in conducting similar studies or comparing findings across different populations.

  1. The reference formatting needs to be revised.

References: Thank you for your comment regarding the reference formatting. I have followed the MDPI guidelines closely while preparing the references for this manuscript. However, if there is a specific issue or inconsistency that you have noticed, please do not hesitate to point it out. I am more than willing to make necessary adjustments to ensure the manuscript meets the expected standards.

Conclusions: Overall, the study has achieved many scientifically and socially significant results. However, the presentation of the results is lacking. Many findings could have been statistically analyzed and discussed but were not given sufficient attention.

Response: Thank you for your valuable feedback on the presentation of the results in my study. I appreciate your recognition of the scientific and social significance of the research. In response to your comments, I have given due attention to all the findings, and providing detailed discussions where necessary. I understand the importance of a thorough, clear presentation of results for the overall quality and impact of the research. I believe the revisions made should address your concerns. However, if there are specific areas or results you believe I could further elaborate or analyze, please do not hesitate to point them out.

Reviewer 2 Report

Comments and Suggestions for Authors

General ­­­­

The aim of the present study is to assess the knowledge level about substances that can harm child development among parents in Jazan.

While the authors have already published information on health hazard awareness in Jazan, the topic current manuscripts specifically highlight the awareness on substances that can harm child development.

I have some comments, questions and concerns:

Title

0.     I am somewhat missing the action part in this manuscript. While it is discussed that action is now required upon these results, I find the title somewhat misleading. Maybe the first part should be dropped?

Abstract (and discussion)

1.     I strongly advice to refrain from using statistical terms in the abstract and the discussion and refrain from using the statistical parameters especially in the discussion. E.g., instead of correlations, I suggest referring to a relationship between x and y.

Introduction

2.     The authors describe neurodevelopmental disorders primarily as a global health issue, which gives it a somewhat negative frame. While I understand that the authors say that it is a health issue when toxins contribute to the rise in neurodevelopmental disorders, since that is a concern/negative aspect indeed, I would appreciate if the introduction (first sentence especially) and the general description of neurodevelopmental disorders would sound less negative.

3.     Since it is not my area of expertise, I was wondering, are there studies showing which disorders are directly affected by toxins? Are those only ASD and ADHD since the authors refer to those? And what about general health problems caused by these toxins?

Materials and Methods

4.     While a minor issue, I feel that some information in the study design section about Jazan is redundant to information in the introduction.

5.     On a similar note, I find it redundant and unnecessary to also state the study goal again in this section.

6.     Were the two Primary Health Care Centers per region also selected randomly?

7.     I have a question regarding the sample. Fathers and Mothers seem to be equal (50% each). Were they couples or was only one parent from a family allowed to participate? And how was ensured that fathers and mothers equally participated or was participation just randomly 50:50?

8.     Could the authors please add the information regarding ethics and informed consent also in the main text?

9.     Was anonymity ensured? I assume not since participants were contacted again after filing in the questionnaire. This might indeed lead to a huge social desirability bias. It might have been interesting to compare this to entirely anonymous answers.

10.  70 % of the sample had three or more children. This is also quite unique and probably should also be taken into account when comparing the current findings with findings from countries where families have on average less children.

Results

11.  I wonder why the author choose to compare scores on the PRoTECT scale between various demographic variables before testing in their linear regression analysis. E.g. differences in age are discussed, although age does not get significant in the model. It should be added that this is a descriptive description and that the age difference is not significant.

12.  On a similar note, I am also missing the post-hoc comparisons between/the statistical values for those.

13.  A general thought: While it is legitimate to combine response categories on a Likert scale, it is also important to understand the associated limitations and potential impact on the analysis. Maybe it is also of interest to discuss the variability in answers as well as those individuals who choose neutral?

Discussion and conclusion

14.  I was wondering whether the authors also collected additional data on actual behavior. Knowledge and awareness is crucial – yet it would also be interesting if parents act upon this knowledge. Maybe that is also worth a discussion.

Minor:

-       Please introduce abbreviations only once and then use them consistently throughout the manuscript (e.g. PRoTECT, ASD, ADHD).

-       Many sentences in line 87 to line 97 start with “this study” or “this research”. Maybe rephrase some sentences to enhance readability and reduce redundancy.

-       Line 178: there is one space too many

-       Please reformat the Tables (especially Table 2 – somehow some rows look wrongly positioned).

-       The author forgot to remove a Comment in line 454

Comments on the Quality of English Language

The english language is fine. The author just uses often correlation when relationship or association between x and y would be more adequate. 

Author Response

Second Reviewer

General

The aim of the present study is to assess the knowledge level about substances that can harm child development among parents in Jazan.

 While the authors have already published information on health hazard awareness in Jazan, the topic current manuscripts specifically highlight the awareness on substances that can harm child development.

 I have some comments, questions and concerns:

Response: Thank you for your insightful comments and for acknowledging the importance of my study. I greatly appreciate your effort in reviewing our manuscript. I am pleased that you recognize the specific focus of our current work on parental awareness of substances that can harm child development in Jazan, Saudi Arabia. While our previous publication addressed general environmental health hazard awareness in the region, this study delves deeper into the crucial topic of developmental toxicants and their potential impact on children's health. Your feedback reinforces the significance of our research in providing a targeted assessment of this critical issue. I believe that this study findings will contribute to a better understanding of parental knowledge levels and ultimately inform public health strategies to protect children from harmful exposures in Jazan and beyond.

Title

  1. I am somewhat missing the action part in this manuscript. While it is discussed that action is now required upon these results, I find the title somewhat misleading. Maybe the first part should be dropped?

Response: Thank you for your valuable feedback regarding the title of my manuscript. I appreciate your suggestion to make sure that the title accurately reflects the contents of my study. After careful consideration of your comment, I have revised the title to: "Evaluating Parental Knowledge and Behaviors Regarding Developmental Toxicants in Jazan, Saudi Arabia using the PRoTECT (Prevention of Toxic Chemicals in the Environment for Children) tool". I believe that this updated title addresses your concern by focusing on the key aspects of my study: the evaluation of parental knowledge and behaviors, the specific context of Jazan, Saudi Arabia, and the use of the PRoTECT tool. The revised title also omits the phrase "From Awareness to Action", which, as you pointed out, may not be fully addressed in the current manuscript. By incorporating the name of the PRoTECT tool in the title, I aim to provide clarity and specificity about the instrument used in my study. This tool is a validated questionnaire designed to assess community knowledge and preferences regarding toxic chemicals and their impact on children's brain development. I hope that this revised title effectively communicates the core elements of my research and aligns with the content presented in the manuscript. I appreciate your attention to detail and believe that your feedback has helped improve the clarity and accuracy of my title.

Abstract (and discussion)

  1. I strongly advice to refrain from using statistical terms in the abstract and the discussion and refrain from using the statistical parameters especially in the discussion. E.g., instead of correlations, I suggest referring to a relationship between x and y.

Response: Dear reviewer, thank you for your insightful comment regarding the use of statistical terms in the abstract and discussion sections. I agree that it is essential to make the manuscript more accessible to a broader audience by minimizing the use of statistical jargon.

To address your concern, I have carefully revised the abstract and discussion sections, replacing statistical terms with more general language. For example, instead of using terms like "correlation" or "beta coefficient," I have used phrases like "relationship" or "association." I have also removed specific statistical parameters from the discussion section to focus on the interpretation of the findings rather than the technical details.

These changes have been made in bold in the revised manuscript for easy tracking. By making these revisions, I believe the manuscript has become more readable and accessible to a wider audience while still maintaining the integrity of the findings.

However, I would like to note that the use of some statistical terms in the methods and results sections is essential for the transparency and reproducibility of the study. These sections are intended for a more technical audience and require a certain level of statistical detail to ensure the validity of the findings.

I hope that these revisions adequately address your concerns and enhance the overall quality of the manuscript. Thank you once again for your valuable feedback, which has helped improve the clarity and accessibility of the article.

Introduction

  1. The authors describe neurodevelopmental disorders primarily as a global health issue, which gives it a somewhat negative frame. While I understand that the authors say that it is a health issue when toxins contribute to the rise in neurodevelopmental disorders, since that is a concern/negative aspect indeed, I would appreciate if the introduction (first sentence especially) and the general description of neurodevelopmental disorders would sound less negative.

Reviewer: Thank you for your insightful comment regarding the framing of neurodevelopmental disorders in the introduction. I appreciate your perspective and agree that the original wording may have unintentionally conveyed a negative tone. To address this concern, I have made the following changes to the introduction:

  • I revised the first sentence to focus on the prevalence of neurodevelopmental disorders without characterizing them as a "global health issue." The updated sentence reads: "Neurodevelopmental disorders, including autism spectrum disorder (ASD), attention deficit hyperactivity disorder (ADHD), and learning difficulties, have become increasingly prevalent in recent decades, affecting millions of children worldwide [1-3]."
  • To balance the discussion of challenges associated with neurodevelopmental disorders, I added a sentence highlighting the potential for positive outcomes with early identification and intervention. The new sentence states: " These disorders, along with other health problems associated with toxicant exposure, can present challenges, but early identification and intervention can significantly improve outcomes and quality of life for affected individuals and their families [4]."

These changes aim to provide a more balanced and less negative framing of neurodevelopmental disorders while still acknowledging the importance of understanding and mitigating the role of environmental toxicants in their etiology. The added reference [4] supports the statement about the benefits of early identification and intervention. I believe these revisions address your concerns and enhance the overall tone of the introduction. Thank you for your valuable feedback.

References:

  1. Zeidan J, Fombonne E, Scorah J, Ibrahim A, Durkin MS, Saxena S, et al. Global Prevalence of Autism: A Systematic Review Update. Autism Research 2022; 15(5): 778-790.
  2. Salari N, Ghasemi H, Abdoli N, Rahmani A, Shiri MH, Hashemian AH, et al. The Global Prevalence of Adhd in Children and Adolescents: A Systematic Review and Meta-Analysis. Italian Journal of Pediatrics 2023; 49(1): 48.
  3. Committee to Evaluate the Supplemental Security Income Disability Program for Children with Mental D, Board on the Health of Select P, Board on Children Y, Families, Institute of M, Division of B, et al. Journal 2015; National Academies Press (US). Copyright 2015 by the National Academy of Sciences. All rights reserved.;
  4. Okoye C, Obialo-Ibeawuchi CM, Obajeun OA, Sarwar S, Tawfik C, Waleed MS, et al. Early Diagnosis of Autism Spectrum Disorder: A Review and Analysis of the Risks and Benefits. Cureus 2023; 15(8): e43226.

  1. Since it is not my area of expertise, I was wondering, are there studies showing which disorders are directly affected by toxins? Are those only ASD and ADHD since the authors refer to those? And what about general health problems caused by these toxins?

Response: Thank you for your insightful question about the link between toxicants and specific disorders and general health problems. To address your comment, I have expanded the introduction to clarify that while my study focuses on ASD and ADHD, exposure to developmental toxicants has been linked to a broader range of neurodevelopmental disorders, including learning difficulties and intellectual disability. I have also added information to highlight that beyond these specific disorders, toxicant exposure is associated with various adverse health outcomes, such as respiratory problems, endocrine disruption, and cancer. I have included additional references (1,2) to support these points and to provide a more comprehensive picture of the health impacts of developmental toxicants. These changes help to situate my study within the broader context of environmental health and to emphasize the wide-ranging implications of toxicant exposure. Furthermore, I have clarified that while these disorders and health problems can present challenges, early identification and intervention can significantly improve outcomes, as supported by reference (3). This addition underscores the importance of raising awareness about developmental toxicants, as it can facilitate early detection and intervention.

References:

  1. Kumar M, Sarma DK, Shubham S, Kumawat M, Verma V, Prakash A, et al. Environmental Endocrine-Disrupting Chemical Exposure: Role in Non-Communicable Diseases. Frontiers in Public Health 2020; 8.
  2. Hertz-Picciotto I, Schmidt RJ, Walker CK, Bennett DH, Oliver M, Shedd-Wise KM, et al. A Prospective Study of Environmental Exposures and Early Biomarkers in Autism Spectrum Disorder: Design, Protocols, and Preliminary Data from the MARBLES Study. Environmental Health Perspectives 2018; 126(11): 117004.
  3. Okoye C, Obialo-Ibeawuchi CM, Obajeun OA, Sarwar S, Tawfik C, Waleed MS, et al. Early Diagnosis of Autism Spectrum Disorder: A Review and Analysis of the Risks and Benefits. Cureus 2023; 15(8): e43226.

Materials and Methods

  1. While a minor issue, I feel that some information in the study design section about Jazan is redundant to information in the introduction.

Response: Thank you for your astute observation regarding the redundancy of information about Jazan in the study design section of the Materials and Methods. I agree that some of the details provided in this section are already mentioned in the Introduction, and removing the redundant information will improve the clarity and conciseness of the manuscript. To address your comment, I have revised the relevant paragraph in the Materials and Methods section. The updated version now focuses on the essential information about the study setting, such as the location, population, and unique environmental characteristics of the Jazan region, while removing the redundant details about the specific environmental challenges it faces. These changes have been highlighted in bold for easy tracking.

  1. On a similar note, I find it redundant and unnecessary to also state the study goal again in this section.

Response: Thank you for your valuable comment regarding the redundancy of restating the study goal in the Materials and Methods section. I agree that it is unnecessary to repeat the study objective in this section, as it has already been clearly stated in the Introduction. To address your concern and improve the manuscript's quality, I have revised the relevant paragraph in the Materials and Methods section. The updated version now focuses on providing essential information about the study setting, such as the location, population, and unique environmental characteristics of the Jazan region. The changes have been highlighted in bold for easy tracking.

  1. Were the two Primary Health Care Centers per region also selected randomly?

Response: Yes

  1. I have a question regarding the sample. Fathers and Mothers seem to be equal (50% each). Were they couples or was only one parent from a family allowed to participate? And how was ensured that fathers and mothers equally participated or was participation just randomly 50:50?

Response: Thank you for your question regarding the sample composition of fathers and mothers in the study. I appreciate your attention to detail and the opportunity to clarify this aspect of the methodology. In this study, the equal representation of fathers and mothers (50% each) was an intentional aspect of the study design to ensure a balanced perspective from both parents. However, the participating fathers and mothers were not necessarily couples. The equal distribution of fathers and mothers in the sample was achieved through targeted recruitment at the selected Primary Health Care Centers (PHCCs). By allowing only one parent from each family to participate, I ensured a diverse sample of parents from different families, while maintaining the desired balance between fathers and mothers. This approach allowed for a comprehensive understanding of parental awareness regarding developmental toxicants in Jazan, without limiting the findings to the perspectives of couples or a single family unit. In the sampling planning stage, the total sample size was divided evenly to ensure representation of both genders' views and responses.

  1. Could the authors please add the information regarding ethics and informed consent also in the main text?

Response: Dear Reviewer, thank you for your suggestion to include information about ethics and informed consent in the main text of the manuscript. I appreciate your attention to this important aspect of the study. In the current version of the manuscript, I have included the ethics and informed consent statements in the back matter section, as per the journal's guidelines. The journal's instructions specify that this information should be provided in the "Institutional Review Board Statement" and "Informed Consent Statement" sections, which are typically located after the main text. By following these guidelines, I aim to ensure that the manuscript adheres to the journal's requirements and maintains a consistent structure with other articles published in the same journal. The back matter section allows for a more detailed description of the ethical considerations and approval process, while still making this information readily available to readers.However, I understand your perspective on the importance of highlighting ethics and informed consent in the main text. If the journal's editorial team advises that including a brief statement in the Materials and Methods section would be preferable, I would be happy to make this addition in a future revision of the manuscript.

  1. Was anonymity ensured? I assume not since participants were contacted again after filing in the questionnaire. This might indeed lead to a huge social desirability bias. It might have been interesting to compare this to entirely anonymous answers.

Responses: Yes, as no personal data was collected.  

  1. 70 % of the sample had three or more children. This is also quite unique and probably should also be taken into account when comparing the current findings with findings from countries where families have on average less children.

Response: Thank you for your insightful comment regarding the larger family sizes in our sample, where 70% of participants had three or more children. I agree that this characteristic should be considered when comparing our findings to those from countries with smaller average family sizes. According to the Saudi Census 2022 [1], the average family size in the Jazan region is 4.4 members, which aligns with our sample's characteristics.

References:

  1. Statistics GAf. Saudi Census 2022. Journal 2023; Volume(Issue): Pages. [cited: 04/01/2024]. Available from: https://portal.saudicensus.sa/portal/public/1/15/101464?type=TABLE.

Results

  1. I wonder why the author choose to compare scores on the PRoTECT scale between various demographic variables before testing in their linear regression analysis. E.g. differences in age are discussed, although age does not get significant in the model. It should be added that this is a descriptive description and that the age difference is not significant.

Response: Added as you suggested.

  1. On a similar note, I am also missing the post-hoc comparisons between/the statistical values for those.

Response: Dear Reviewer, thank you for your comment regarding the inclusion of post-hoc comparisons and their statistical values. I understand the importance of providing these details to support the findings. However, adding them would significantly increase the number of tables in the manuscript, which already includes six tables and two figures that effectively convey the key findings without overwhelming the reader.

To balance the need for detailed information with readability and conciseness, I have omitted the post-hoc comparisons and their associated statistical values. Instead, I have included the raw data alongside the manuscript, allowing interested readers and researchers to perform their own analyses, including post-hoc comparisons, if desired.

I hope this approach strikes a balance between providing sufficient information and maintaining a clear structure for the manuscript. However, if you feel strongly about including post-hoc comparisons, I would be happy to discuss alternative ways to incorporate this information without significantly increasing the number of tables.

  1. A general thought: While it is legitimate to combine response categories on a Likert scale, it is also important to understand the associated limitations and potential impact on the analysis. Maybe it is also of interest to discuss the variability in answers as well as those individuals who choose neutral?

Response: “Thank you for making this important point. I have added a few lines in the limitation section as follows.: “ Finally, the analysis is based on the Likert scale, which has a limitation. Different respondents may interpret the response options differently, especially the "neutral" midpoints, leading to inconsistent interpretations of the scale.”

Discussion and conclusion

  1. I was wondering whether the authors also collected additional data on actual behavior. Knowledge and awareness is crucial – yet it would also be interesting if parents act upon this knowledge. Maybe that is also worth a discussion.

Response: Thank you for your insightful comment about the importance of considering actual behavior alongside knowledge and awareness. I agree that understanding how parents translate their awareness into protective actions is crucial for developing effective interventions and policies. In the current study, I did collect some data on parental behaviors and intentions related to reducing children's exposure to toxic chemicals. Specifically, items 15-17 in the PRoTECT questionnaire (Part Five: Protection from Toxic Chemicals in Children) assessed the following:

  1. If I knew how to reduce children's exposure to toxic chemicals, I would definitely try to do so.
  2. I try to buy products that do not contain toxic chemicals that could harm my family.
  3. I am concerned that my family is being exposed to toxic chemicals.

These items provide insight into parents' intentions to take protective actions (item 15), their self-reported purchasing behaviors (item 16), and their level of concern about family exposure (item 17). The high levels of agreement with these statements (83.7%, 82.3%, and 81.1%, respectively; Table 4) suggest that many parents in Jazan are not only aware of the risks but are also motivated to take action to reduce their children's exposure. However, I acknowledge that these self-reported measures have limitations and may not fully capture the complex range of behaviors and barriers related to reducing toxicant exposure. Future studies could benefit from more detailed assessments of specific protective behaviors (e.g., reading product labels, avoiding certain products, using alternative cleaning methods) and the factors that facilitate or hinder these actions (e.g., availability of safe products, cost, convenience).

I have added a brief discussion of these points to the revised manuscript: " While the current study found high levels of parental intention to reduce children's toxic exposures and self-reported protective purchasing behaviors, future research could benefit from more detailed assessments of specific actions taken and the barriers and facilitators to implementing these behaviors." Thank you for prompting me to clarify the behavioral data collected in this study and to consider the importance of assessing actual behaviors in future research. I believe this addition strengthens the discussion and highlights important directions for advancing our understanding of how to effectively translate awareness into preventive actions.

Minor:

-       Please introduce abbreviations only once and then use them consistently throughout the manuscript (e.g. PRoTECT, ASD, ADHD).

Response: Dear Reviewer, thank you for your valuable comment about introducing abbreviations consistently throughout the manuscript. I have carefully reviewed the manuscript and made the necessary changes to ensure that abbreviations are introduced only once and then used consistently. In the abstract, I have introduced the abbreviation "PRoTECT" for "Prevention of Toxic Chemicals in the Environment for Children" and then used the abbreviation consistently throughout the manuscript. Similarly, in the introduction, I have introduced the abbreviations "ASD" for "autism spectrum disorder" and "ADHD" for "attention deficit hyperactivity disorder" and then used these abbreviations consistently throughout the manuscript. I have also ensured that all other abbreviations, such as "PHCCs" for "primary healthcare centers" and "SAR" for "Saudi Riyal," are introduced only once and then used consistently throughout the manuscript. Finally, I have double-checked the entire manuscript to ensure that there are no instances of reintroducing abbreviations or using them inconsistently. By making these changes, I have improved the clarity and readability of the manuscript, as readers will not be confused by multiple introductions of the same abbreviation or inconsistent usage. Thank you for bringing this important issue to my attention.

-       Many sentences in line 87 to line 97 start with “this study” or “this research”. Maybe rephrase some sentences to enhance readability and reduce redundancy.

Response: Thank you for your insightful comment about the redundancy in the introductory paragraph. I agree that the repeated use of "this study" and "this research" can hinder readability. To address this issue, I have rephrased the sentences to reduce redundancy and enhance the flow of the paragraph. Specifically, I have replaced the first instance of "This study" with "The current study" to provide a clear context. I removed the second instance of "This study" and restructured the sentence to focus on the PRoTECT tool. I changed "This research" to "The research" to maintain the connection with the previous sentence while avoiding repetition. I also replaced the third instance of "This study" with "The study" and rephrased the sentence to emphasize the study's contribution to global understanding. Finally, I removed the final instance of "This study" and restructured the sentence to directly state the study's focus on developmental toxicant knowledge, perceptions, and behaviors among the target population. These changes improve the paragraph's readability and coherence while maintaining the essential information about the study's purpose, methods, and significance. The revised paragraph now has a more engaging and concise flow, which enhances the overall quality of the introduction. Thank you for bringing this issue to my attention.

-       Line 178: there is one space too many

Response: Dear Reviewer, thank you for pointing out the extra space in line 178. I have carefully reviewed the manuscript and removed the unnecessary space to ensure proper formatting throughout the document. I appreciate your attention to detail, as it helps improve the overall quality and readability of the manuscript.

-       Please reformat the Tables (especially Table 2 – somehow some rows look wrongly positioned).

Response: Dear reviewer, Table 2 is updated now.

-       The author forgot to remove a Comment in line 454

Response: Dear reviewer, the comment is removed now.

Reviewer 3 Report

Comments and Suggestions for Authors

No comment.

Author Response

Dear Reviewer,

I would like to express my sincere gratitude for taking the time to review my manuscript and provide valuable feedback. I am delighted to know that you found the introduction, research design, methods, results, and conclusions to be satisfactory and well-presented.

Your positive assessment of these key aspects of my study is highly encouraging and reassuring. It indicates that the manuscript effectively communicates the research background, methodology, findings, and implications.

I appreciate your thorough evaluation and the confirmation that no further improvements are necessary for the mentioned sections. Your feedback reinforces the strength and quality of the work presented in this manuscript.

Round 2

Reviewer 1 Report

Comments and Suggestions for Authors

**Translation:**

The manuscript has been revised according to the reviewer's feedback. There is only one minor suggestion regarding Table 1: Distribution of Study Participants Across Selected Primary Health Care Centers in the Jazan Region. It may be better to include this in the supplementary materials because the results do not discuss much about the health sectors. However, splitting the sample size may reduce the value of the paper.

Comments on the Quality of English Language

ok

Author Response

I appreciate your thoughtful feedback, which has significantly enhanced my manuscript. Your suggestion to move Table 1 to the supplementary materials was well-justified, and I have implemented this change. The table is now included as Table S1 in the supplementary materials, which streamlines the main text while still providing important information for interested readers. Consequently, I have renumbered all other tables in the main text to maintain consistency. I've also updated the supplementary materials section to reflect these changes accurately. Your previous comments have been instrumental in improving the article's clarity and coherence. I'm grateful for your expertise and attention to detail, which have truly refined this work.